# Identification and Validation of Esophageal Squamous Cell Carcinoma Targets for Fluorescence Molecular Endoscopy

**DOI:** 10.3390/ijms22179270

**Published:** 2021-08-27

**Authors:** Xiaojuan Zhao, Qingfeng Huang, Marjory Koller, Matthijs D. Linssen, Wouter T. R. Hooghiemstra, Steven J. de Jongh, Marcel A. T. M. van Vugt, Rudolf S. N. Fehrmann, Enmin Li, Wouter B. Nagengast

**Affiliations:** 1Department of Medical Oncology, University Medical Center Groningen, University of Groningen, P.O. Box 30.001, 9700 RB Groningen, The Netherlands; x.zhao01@umcg.nl (X.Z.); m.vugt@umcg.nl (M.A.T.M.v.V.); r.s.n.fehrmann@umcg.nl (R.S.N.F.); 2Department of Gastroenterology and Hepatology, University Medical Center Groningen, University of Groningen, P.O. Box 30.001, 9700 RB Groningen, The Netherlands; m.d.linssen@umcg.nl (M.D.L.); w.t.r.hooghiemstra@umcg.nl (W.T.R.H.); s.j.de.jongh@umcg.nl (S.J.d.J.); 3Guangdong Provincial Key Laboratory of Infectious Diseases and Molecular Immunopathology, The Key Laboratory of Molecular Biology for High Cancer Incidence Coastal Chaoshan Area, Shantou University Medical College, Shantou 515041, China; 17qfhuang@stu.edu.cn (Q.H.); nmli@stu.edu.cn (E.L.); 4Department of Surgery, University Medical Center Groningen, University of Groningen, P.O. Box 30.001, 9700 RB Groningen, The Netherlands; marjorykoller@gmail.com; 5Department of Clinical Pharmacy and Pharmacology, University Medical Center Groningen, University of Groningen, P.O. Box 30.001, 9700 RB Groningen, The Netherlands

**Keywords:** fluorescence molecular endoscopy, early detection, squamous high-grade dysplasia, bioinformatics, mRNA profiling

## Abstract

Dysplasia and intramucosal esophageal squamous cell carcinoma (ESCC) frequently go unnoticed with white-light endoscopy and, therefore, progress to invasive tumors. If suitable targets are available, fluorescence molecular endoscopy might be promising to improve early detection. Microarray expression data of patient-derived normal esophagus (*n* = 120) and ESCC samples (*n* = 118) were analyzed by functional genomic mRNA (FGmRNA) profiling to predict target upregulation on protein levels. The predicted top 60 upregulated genes were prioritized based on literature and immunohistochemistry (IHC) validation to select the most promising targets for fluorescent imaging. By IHC, GLUT1 showed significantly higher expression in ESCC tissue (30 patients) compared to the normal esophagus adjacent to the tumor (27 patients) (*p* < 0.001). Ex vivo imaging of GLUT1 with the 2-DG 800CW tracer showed that the mean fluorescence intensity in ESCC (*n* = 17) and high-grade dysplasia (HGD, *n* = 13) is higher (*p* < 0.05) compared to that in low-grade dysplasia (LGD) (*n* = 7) and to the normal esophagus adjacent to the tumor (*n* = 5). The sensitivity and specificity of 2-DG 800CW to detect HGD and ESCC is 80% and 83%, respectively (ROC = 0.85). We identified and validated GLUT1 as a promising molecular imaging target and demonstrated that fluorescent imaging after topical application of 2-DG 800CW can differentiate HGD and ESCC from LGD and normal esophagus.

## 1. Introduction

Esophageal cancer is the sixth leading cause of cancer-related deaths worldwide, resulting in an estimated 508,585 deaths worldwide in 2018 [1]. The two dominant histological subtypes in esophageal cancer are adenocarcinoma and squamous cell carcinoma. Around 87% of global esophageal cancers are of squamous origin. Most of esophageal squamous cell carcinoma (ESCC) were diagnosed in the esophageal cancer belt, which spans from Iran to central-northern China [2]. Despite recent advances in treatment regimens, the 5-year disease-specific survival rate of ESCC remains as low as around 19% [3].

Esophageal squamous dysplasia is the precursor of ESCC. The cumulative ESCC incidence rate for low-grade dysplasia is around 1%. In comparison, the cumulative ESCC incidence rate is 6% for high-grade dysplasia (HGD, *p* Tis N0 M0) [4]. After tumor invasion of the muscularis mucosa, the overall risk of metastasis increases and the 5-year overall survival rate of ESCC patients decreases dramatically [5,6]. If the tumor is limited to epithelial (EP) and the lamina propria (LPM) layer of intramucosal carcinoma (*p* Tis-T1aEP/LPM N0 M0) [7,8,9], endoscopic resection offers a minimally invasive option for high-grade dysplasia and ESCC patients, which improves quality of life and increases the 5-year overall survival rate to around 90% [5].

Nevertheless, around 70% of ESCC patients are diagnosed with stage II to IV carcinoma [3]. Chromoendoscopy with Lugol’s staining in combination with high-definition white light (HDWL) imaging is considered the best available technique for detecting squamous HGD and ESCC [10]. The Lugol’s staining shows brownish-black staining in normal squamous epithelium containing high levels of glycogen but shows less intense or absent staining in squamous HGD or carcinoma as these cells contain little glycogen [10]. A large cohort analysis involving 586 participants for Lugol chromoendoscopy screening reports a detection sensitivity of around 71% for squamous HGD [11]. Lugol’s staining could also lead to retrosternal chest pain and some complications, such as hypersensitivity to iodine, esophagitis, laryngitis, and bronchopneumonia [12]. These complications and the low sensitivity limit the application of Lugol’s staining. A recent randomized controlled trial revealed that HDWL endoscopy combined with narrow-band imaging (NBI) system missed all esophageal squamous dysplasia lesions (LGD 0/5; HGD 0/2) [13]. These facts highlight the necessity of improving endoscopic methods to increase detection of high-grade dysplasia and ESCC.

The progression of dysplasia and ESCC from native squamous epithelium is characterized by genomic alterations, such as driver mutations in tumor suppressor genes *NOTCH1* and *TP53*, oncogene *NFE2L2*, activation of NRF2 pathway and WNT-β-catenin pathway [14]. Some of the genomic alterations that frequently occur in dysplasia and carcinoma lead to downstream protein overexpression. Fluorescence molecular endoscopy (FME) can target and fluorescently highlight the overexpressed protein targets in dysplasia and carcinoma, which might improve the sensitivity of early detection. FME targeting of excreted factors, such as vascular endothelial growth factor (VEGF) has already been tested in clinical trials on esophageal cancer [15], which demonstrated the safety and feasibility of this technique. However, VEGF is only upregulated in 31% to 69% of ESCC lesions and, therefore, is most likely not sensitive enough [16,17]. In addition, VEGF is excreted by cells and is mainly located in the stroma which could lead to a false-positive in vivo detection. This restriction could be resolved by using a membrane-bound antigen because FME would target the tumor cell itself. Therefore, in this study, we aim to detect novel imaging protein targets which are membrane-bound and have the potential to improve the detection of esophageal squamous HGD and ESCC lesions by FME.

## 2. Results

### 2.1. Functional Genomic mRNA Profiling Identifies Upregulated Genes in Esophageal Squamous Cell Carcinoma

We performed a FGmRNA profiling with publicly available expression profiles of 118 ESCC samples and 120 normal esophagus samples (Appendix A). Class comparison analysis with multivariate permutation testing identified 5364 significantly upregulated genes in ESCC compared to normal esophagus samples (Appendix A).

### 2.2. Target Prioritization by Literature Search

The cellular localization of protein products of the top 60 genes is shown in Figure 1. An overview of the literature search results on the top 60 upregulated genes is presented in Appendix A.

The literature search was performed on the first 60 upregulated genes. Among these 60 genes (Appendix A), 44 genes are described to play a role in carcinogenesis. 12 genes have a known downstream protein overexpression in human esophageal cancer samples. The protein products of 20 genes have shown overexpression in gastrointestinal tumors and these genes could be of interest for ESCC. The encoding proteins of 12 genes are predicted with the cellular location on plasma membrane.

An imaging target for distinguishing ESCC from a normal esophagus should meet two requirements. First, it is preferably localized on the cell membrane, which is more easily accessible for monoclonal antibodies or peptides than intracellular targets. Secondly, the protein should be related to tumorigenesis and have been reported to be overexpressed in esophageal cancer or gastrointestinal cancer compared to normal tissue. According to these requirements, five proteins were defined as possible targets, including FZD6 (Rank 12), GLUT1 (Rank 27), ENTPD1 (Rank 36), LEPR (Rank 43), and IFNGR1 (Rank 48).

### 2.3. Evaluation of Five Possible Targets by IHC

As there is no anti-FZD6 monoclonal antibody commercially available for IHC validation and no fluorescent tracer is available, we at first excluded FZD6 from further validation. We performed IHC of ENTPD1, IFNGR1, LEPR, and GLUT1 on patient’s tissue including both ESCC (*n* = 5) and normal esophagus (*n* = 5). Anti-ENTPD1 IHC showed stronger positive staining in stroma cells surrounding esophageal squamous cancer cells compared to normal squamous epithelium cells with localization on cell membrane. However, there was no expression of ENTPD1 detected, neither on the normal esophagus epithelium cells nor on the squamous epithelium cancer cells. Anti-LEPR IHC showed weak positivity in normal squamous epithelial cells but much stronger positivity in esophageal squamous cancer cells. However, we tested three anti-LEPR antibodies and they only showed cytoplasm staining, with no cell membrane staining (Figure 2). Anti-IFNGR1 positive staining is found in ESCC tissue but not in normal squamous epithelium. However, after testing two monoclonal anti-IFNGR1 antibodies, we found that both of them show only cytoplasmic staining. Anti-GLUT1 IHC showed that GLUT1 has a high expression on the cell membrane of ESCC with low staining of normal squamous epithelium (Figure 2). Therefore, we decided to expand the cohort to validate GLUT1 expression by IHC with TMA.

### 2.4. Ex Vivo Validation of GLUT1 by IHC on TMA: GLUT1 as a Potential Target in ESCC Populations

To validate if GLUT1 can serve as an imaging target to distinguish ESCC from normal esophagus, we analyzed GLUT1 protein expression in both ESCC and normal esophagus from the same patient by IHC on tissue microarray (Figure 3). GLUT1 showed dominant cell membrane staining in esophageal squamous epithelial cancer cells. For GLUT1, 83.33% (25/30) of ESCC samples show intermediate to high positive intensity, only 16.67% (5/30) of ESCC tissue show low to negative positivity, compared to normal esophagus tissue, which shows 100% (27/27) low-to-negative GLUT1 expression (*p* < 0.001).

### 2.5. GLUT1-Related Fluorescence Imaging (2-DG 800CW) of Endoscopy Biopsies Ex Vivo

The NIR fluorescent glucose analogue, 2-DG 800CW, is a fluorescent marker whose uptake is mediated by GLUT1 [18]. To investigate the uptake in HGD and ESCC, we collected biopsies from patients undergoing surveillance endoscopies (*n* = 29) and sprayed the tracer 2-DG 800CW ex vivo and calculated the mean fluorescence intensity (MFI) in each individual biopsy. The MFI in ESCC (*n* = 17, 4309 ± 2657) tissue showed a significant difference compared to that in low-grade dysplasia (LGD, *n* = 7, 2027 ± 1190) and normal esophagus (*n* = 5, 1323 ± 589.7) (*p* < 0.05). In addition, the MFI in high-grade dysplasia (HGD, *n* = 13, 4596 ± 2135) shows a significant difference compared to that in LGD and normal esophagus (*p* < 0.01) There is no significant difference between the MFI for LGD versus normal esophagus (*p* = 0.2677) and HGD versus ESCC (*p* = 0.7378) (Figure 4a). To determine the sensitivity and specificity of 2-DG 800CW to detect HGD and ESCC by MFI, a ROC curve was generated which showed an area of 0.8528 (Figure 4b). If differentiating HGD and ESCC from a normal esophagus with a cut-off MFI value of 2355, the sensitivity and specificity are 80% (24/30) and 83.33% (10/12), respectively. To further correlate the fluorescent signal to histology accurately, we compared the 2-DG 800CW signal to the H&E on 4 μm slices. We found that most of 2-DG 800CW fluorescence was located on tumor cells (Figure 5a,b), while some fluorescence was also found to be along the edges of the biopsy (see Figure 5c,d). Next, we compared the 2-DG 800CW signal to the GLUT1 IHC results and tumor area on 4 μm slices and indeed found that the fluorescent signal was not present in all GLUT1 positive areas on the 4 μm slices (see Figure 5a,b). We calculated the sensitivity (the percentage of 2-DG 800CW positive area in GLUT1 positive area) and specificity (the percentage of 2-DG 800CW negative area in GLUT1 negative area) of 2-DG 800CW in detection of GLUT1 overexpression on 4 μm slices of HGD and ESCC biopsies, resulting in a median sensitivity of 23.02% and a median specificity of 85.43% (*n* = 19) (see Figure 4c). We also calculated the sensitivity and specificity of 2-DG 800CW in detection of HGD and ESCC on 4 μm slices, resulting in a median sensitivity of 18.76% and a median specificity of 74.18% (*n* = 19) (see Figure 4c).

## 3. Discussion

Esophageal squamous cell carcinoma is reported with a high incidence rate and poor prognosis [2,3]. Its precursor stage of high-grade dysplasia shows a high risk of progression to ESCC [4]. Fluorescence molecular endoscopy has the potential to improve the detection of HGD and ESCC by targeting tumor-specific molecules at protein level. By using FGmRNA profiling, we identified novel targets for detection of dysplasia and mucosal ESCC. Of these, GLUT1 showed most promising for molecular imaging and demonstrated that fluorescent imaging after topical application of 2-DG 800CW can differentiate HGD and ESCC from LGD and normal esophagus tissue.

Previously, we performed FGmRNA profiling in colorectal cancer and showed that FGmRNA profiling can obtain useful information from pre-existing data and find a proper new target for molecular imaging [19]. Somatic copy number alterations (SCNAs) participate in the stepwise progression from native squamous epithelium to squamous dysplasia and ESCC [14]. However, the effect of SCNAs on gene expression levels is often overshadowed by non-genetic factors, such as physiological circadian rhythm or metabolic factors [20]. The non-genetic transcriptional components were applied as covariates to correct the raw microarray expression data, namely, FGmRNA profiling. The residual gene expression levels (FGmRNA profiles) after correction correlation with SCNAs capture the downstream effect of SCNAs on gene expression levels. In this study, among our top identified genes, *DVL2*, *DVL3*, *FZD6*, and *GSK3B* are all known target-genes of the Wnt signaling pathway. In addition, *ADAM17*, *ELF3*, and *EGFR* belong to the Notch signaling pathway. The Wnt signaling pathway and the Notch pathway are both known to be involved in the carcinogenesis of ESCC [21,22], which supports the pathophysiological relevance of the identified genes. Trophoblast cell surface antigen-2 (Trop-2) contributes to carcinogenesis and is overexpressed in several cancer types [23]. Nakashima K, et al. reported an overexpression of Trop-2 in ESCC tissue by immunohistochemistry [24], in concordance with our FGmRNA profiles (Rank 335). Sacituzumab govitecan, targeting Trop-2, was developed [25] and under investigation in breast cancer (NCT02574455, NCT03901339) and urothelial cancer (NCT04527991). We present an extensive list of additional genes identified by FGmRNA profiling, which may be the new potential imaging targets (Appendix A).

After validation of four proteins by IHC, we find that ENTPD1, IFNGR1, LEPR, and GLUT1 show higher expression in ESCC compared to the normal esophagus, which is consistent with FGmRNA profiling results. ENTPD1 was excluded as an FME target as we saw it was expressed in the tumor microenvironment. This is expected as ENTPD1 is an integral component of regulatory T cells, which infiltrate into the tumor stroma. As the differential expression of ENTPD1 is not on epithelial cells, we do not consider ENTPD1 as an ideal target for imaging. According to the literature search, LEPR and IFNGR1 are described to express mostly on the cell membrane, sometimes in cytoplasm. With the antibodies we used, membrane staining was not shown.

GLUT1 is a member of glucose transporter (GLUT) family, which can transport glucose and related hexoses into cancer cells to meet their high metabolic requirement [26]. To date, there are 14 members of the GLUT family reported, among which GLUT1 [27], GLUT2 [28], GLUT3 [27], and GLUT4 [29] have a role in ^18^FDG uptake. The transporters of ^18^FDG are still controversial and which specific GLUT subtypes are in charge of FDG uptake differs between cancer types. The facts that both the expression of different GLUT proteins and the sensitivity of ^18^FDG–PET varies greatly between different cancer types [26,30], which suggests the molecular imaging of specific GLUT members can improve the detection sensitivity of different cancer types.

IRDye 800CW conjugated to 2-DG is a fluorescently labeled glucose-derivative tracer with a molecular weight of around 1330 Da, which is more than seven-folds higher than that of ^18^FDG (around 181 Da). The high uptake of 2-DG 800CW in cancer is thought to be because of the high metabolic requirement, which is the same as ^18^FDG. Although 2-DG 800CW is also proven to be transported into cells, its transportation mechanism is still unknown. It is suggested to result from an endocytosis process of GLUT1/2-DG 800CW complexes and it is also shown to be related to GLUT4 [18]. Our results have shown that the uptake of 2-DG 800CW in HGD and ESCC is significantly higher than in normal squamous epithelium or low-grade dysplasia. Previous reports have shown 2-DG 800CW uptake in breast cancer and prostate carcinoma in mouse models [18,31], while we show its potential role after topical administration in HGD and ESCC cancer with patients-derived endoscopic biopsies. Furthermore, we find that by differentiating HGD and ESCC from normal esophagus tissue at a cut-off MFI value of 2355, the sensitivity and specificity is 80% and 83.33%, respectively. However, when comparing the fluorescent signal to GLUT1 expression or tumor area on 4 μm slices, we find that 2-DG 800CW is highly specific to detect GLUT1 or a tumor, but the sensitivity is relatively low. We think there are some limitations leading to the low sensitivity on 4 μm slices. First, the tracer is sprayed ex vivo on the biopsies but not injected in vivo. Because of the tight junctions between the surface epithelium cells of the esophageal mucosal layer, it is not easy for the tracer to reach cells expressing membrane-bound GLUT1. Second, the incubation time may not be long enough for all tracers to bind to GLUT1. Furthermore, 2-DG 800CW is not directly binding to the cell surface site of GLUT1 but it is potentially transported into the cytoplasm by endocytosis [18], which requires the cells to be alive. Therefore, as the tracer is tested ex vivo, a reduction in cell viability may be the cause of reduced tracer uptake.

From our results, we can see that 2-DG 800CW is a promising tracer after topical application to the mucosa to distinguish high-grade dysplasia and esophageal squamous cell carcinoma from normal squamous epithelium and low-grade dysplasia with a good sensitivity and specificity. However, ESCC was reported with intertumor and intratumor heterogeneity [32,33]. Recently, the first multiplexed FME system was developed, which would enable concurrent imaging of multiple target molecules [34]. Further development of other specific fluorescence tracers for the multiplexed imaging with 2-DG 800CW has the potential to further improve the detection of HGD and mucosal ESCC by fluorescence molecular endoscopy.

## 4. Materials and Methods

### 4.1. Identification of Differentially Expressed Genes with Functional Genomic mRNA Profiling

Microarray expression data of primary patient-derived ESCC and normal esophagus samples generated with the Affymetrix HG-U133 plus 2.0 and the HG-U133A platforms were obtained from the Gene Expression Omnibus database [35]. Pre-processing, normalization, and quality control was performed as previously described [36]. Subsequently, functional genomic mRNA (FGmRNA) profiling was applied to extract the downstream consequences of genetic alterations on gene expression profiles [19]. FGmRNA profiling is a method capable of correcting gene expression data and allows for an enhanced view on the downstream effects of genomic alterations on gene expression levels. To identify genes which are upregulated in ESCC compared to the normal esophagus, we performed a transcriptome-wide class comparison (Welch’s *t*-test) between FGmRNA profiles of ESCC and normal esophagus [37,38,39,40,41,42,43,44].

### 4.2. Prioritization Strategy

Based on the FGmRNA results, the top 60 upregulated genes in ESCC were selected for further prioritization based on a literature search [45,46,47,48,49,50,51,52,53,54,55,56,57,58,59,60,61,62,63,64,65,66,67,68,69,70,71,72,73,74,75,76,77,78,79,80,81,82,83,84,85,86,87,88,89,90,91,92,93,94,95,96,97,98,99,100,101,102,103,104,105,106,107,108,109,110,111,112,113,114,115,116,117,118,119,120,121,122,123,124,125]. First, we searched for the function of the gene and its relation to carcinogenesis. Second, we searched per gene for articles published in English from conception until December 2019. The searching terms used are: HUGO gene symbol of the target under investigation in combination with ‘immunohistochemistry’ and ‘esophageal cancer’ or ‘gastrointestinal cancer’ or ‘gastric cancer’ or ‘colorectal cancer’ or ‘rectum cancer’. Thirdly, the cellular localization, function, and expression of the protein product was searched in www.proteinatlas.org, www.Genecards.com and www.genetica-network.com (accessed between 1 January 2017 and 31 December 2019) [126]. As we were looking for promising targets for fluorescent molecular endoscopy, the targets should be expressed on cell membrane to be easily accessible. Furthermore, we evaluated if there was already optical molecular imaging research targeting the protein products of the upregulated genes.

### 4.3. Immunohistochemistry

To validate the expression of protein products encoded by upregulated genes, we performed immunohistochemistry (IHC) on tissue microarrays (TMA) made from formalin-fixed and paraffin-embedded (FFPE) tissue blocks. The tissue blocks include esophageal squamous cell carcinoma (ESCC, *n* = 30) and normal esophagus tissue adjacent to ESCC (*n* = 27) collected from patients who underwent esophagectomy between 2012 and 2016 in the affiliated cancer hospital of Shantou University Medical College (SUMC, Shantou, China). We also performed IHC on FFPE slices including normal esophagus (*n* = 5) and ESCC tissue (*n* = 5) from the University Medical Center Groningen (UMCG). All human tissue samples were applied according to the guidelines of Shantou University Medical College (SUMC) ethics board and that of the University Medical Center Groningen (UMCG) ethics board (www.ccmo.nl, accessed on 22 December 2016). Five-micron slices (for TMA) and four-micron slices (for normal FFPE slices) were cut and mounted on amino-propylethyoxy-silan-coated glass slides. Secondly, the slices were deparaffinized in xylene and rehydrated in 70%, 96%, 99.9% alcohol, respectively. Thirdly, after conducting heat-induced antigen retrieval using citrate buffer (10 mM, pH 6.0) or EDTA buffer (mM, pH 9.0), we blocked the endogenous peroxidase activity by incubation with a 0.3% hydrogen–peroxide solution for 30 min. Next, we incubated the slides with the primary antibodies against GLUT1, ENTPD1, LEPR overnight at 4 degrees (GLUT1) or for 1 h at room temperature (ENTPD1, LEPR, IFNGR1) (Appendix A). Subsequently, the slices were incubated with the second and third antibodies diluted as 1:50 in PBS with 1% human AB-serum and 1% bovine serum albumin (BSA). The incubation lasts for 30 min with either the second or third antibody. Finally, the slices were specifically stained with 3,3′diaminobenzidine (DAB) substrate chromogen solution or 3-amino-9-ethylcarbazole (AEC) peroxidase substrate solution, followed by hematoxylin nuclei staining. XZ optimized the IHC staining protocols, performed the IHC staining under the supervision of a dedicated gastrointestinal pathologist Dr. Arend Karrenbeld (AK) from the Department of Pathology and Medical Biology, the University Medical Center Groningen. Both positive controls and negative controls (with a specific IgG monoclonal antibody) are conducted in each IHC staining to ascertain specific binding of the antibody.

### 4.4. H-Score Calculation

We score the staining intensities of the epithelial cells according to a 0–3 scale (Appendix A). The H-Score of each sample was calculated according to the equation (H-Score = 1 * (percentage of cells weakly stained) + 2 * (percentage of cells moderately scored) + 3 * (percentage of cells strongly stained)). The H-Score range is from 0 to 300, leading to three categories of H-Score for each tissue sample (0–100 is defined as a negative/low H-Score; 101–200 = intermediate; 201–300 = high). The H-Scores were independently scored by two researchers (XZ, AK).

### 4.5. GLUT1 Related Fluorescence Imaging Using 2-DG 800CW on Patient-Derived Esophageal Squamous Cell Carcinoma Biopsies

In total, 29 patients and 42 biopsies were included. We collected 17 ESCC biopsies from 15 patients; 13 high-grade dysplasia biopsies from 12 patients; 7 low-grade dysplasia biopsies from 6 patients, and 5 normal biopsies from 5 patients during endoscopy. In 20 patients, one biopsy per patient was included in the study. In 9 patients, more than one biopsy from each patient were included in the study (In 4 patients, 3 biopsies were included from each patient; in 5 patients, 2 biopsies were included from each patient). The study was approved by the Ethics Committee of Cancer Institute and Hospital, Chinese Academy of Medical Sciences (16-171/1250). All participants received oral and written information before informed consent was obtained. The fresh biopsies were incubated for 5 min with a solution of 2-DG 800CW in phosphate buffered saline (Li-COR Biosciences, Lincoln, NE, USA; 0.1 nmol/mL). After rinsing the tissue with normal saline three times, we scanned the tissue with ChemiDoc™ MP scanner (Bio-Rad, Hercules, CA, USA; 835/50 nm; exposure time: 5 s). After that, we cut each biopsy into 4 μm slices and performed H&E staining with a Hamamatsu scanner (Hamamatsu Photonics, Shizuoka, Japan) and 2-DG 800CW imaging with an Odyssey scanner (Li-COR Biosciences, Lincoln, NE, USA; 800 nm; resolution: 21 μm). To analyze the sensitivity and specificity of 2-DG 800CW to detect GLUT1 expression, anti-GLUT1 IHC staining was also performed on the serial 4 μm slices for each biopsy and imaged with a Hamamatsu scanner (Hamamatsu Photonics, Shizuoka, Japan).

### 4.6. Statistical Analyses

We compared the FGmRNA expression levels of healthy control to ESCC patients by the two-sample Student *t*-test. To control for false discovery rate, we performed a multivariate permutation (MVP) test with a false discovery rate (FDR) of 1% and a confidence interval (CI) of 99%. The overexpressing genes were identified and ranked by the level of significance in FGmRNA profiles. The H-Score of tissue samples were analyzed using the non-parametric Kruskal–Wallis test (IBM SPSS 23.0.0; RRID:SCR_002865, IBM Corporation, Armonk, NY, USA). *p*-value < 0.01 was considered significant for H-Score analysis.

## 5. Conclusions

We have shown that our new bioinformatics approach can both identify potential molecular imaging targets and explore the underlying pathophysiological process. We identified and validated that GLUT1 is indeed overexpressed in ESCC and can be targeted by 2-DG 800CW. This approach has the potential to improve detection of high-grade squamous dysplasia and ESCC by FME during screening endoscopy procedures. Phase 1 clinical trials should further elucidate the potential and safety of 2-DG 800CW in detecting high-grade dysplasia and ESCC by FME.

## Figures and Tables

**Figure 1 ijms-22-09270-f001:**
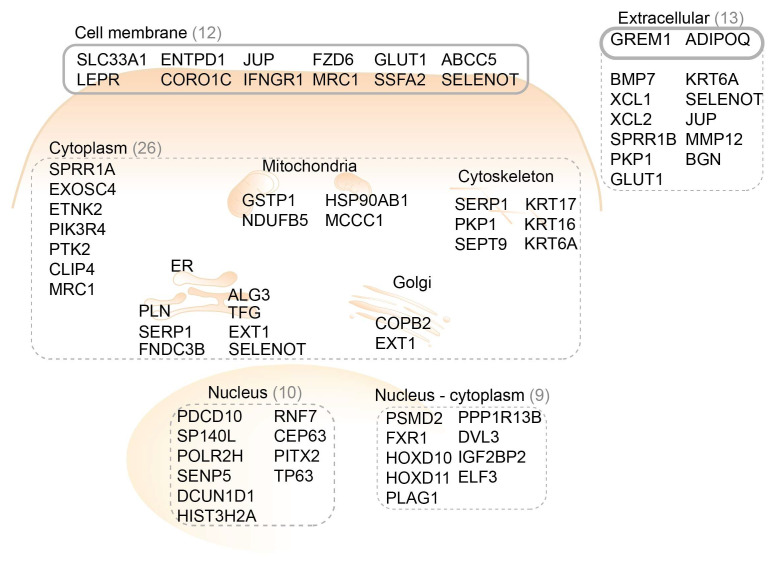
The sub-cellular localization of the protein products of the top 60 FGmRNA-overexpressing genes. The proteins localized on a cell membrane are more easily accessible and especially suitable as imaging targets. ER, endoplasmic reticulum.

**Figure 2 ijms-22-09270-f002:**
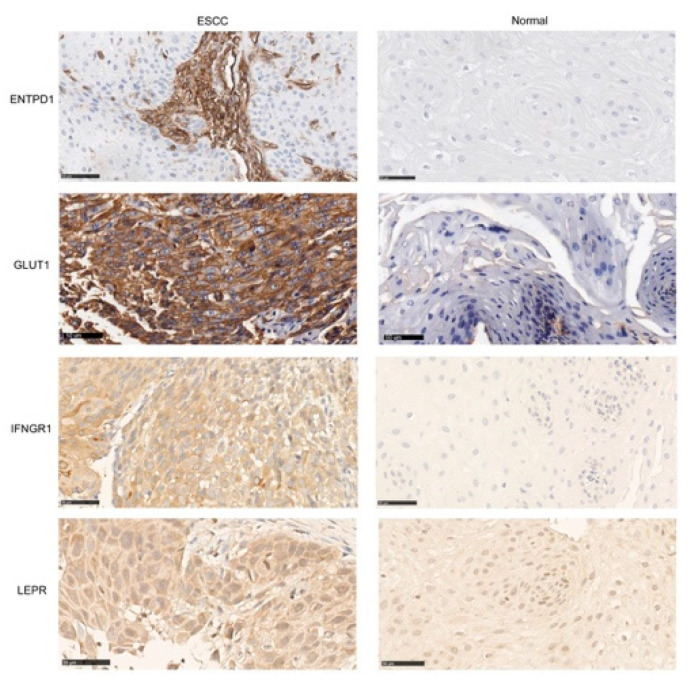
The validation of four candidate genes Representative images of anti-ENTPD1, anti-GLUT1, anti-IFNGR1, anti-LEPR IHC staining results (brown) in ESCC tissue and normal esophagus tissue. Bar: 50 μm.

**Figure 3 ijms-22-09270-f003:**
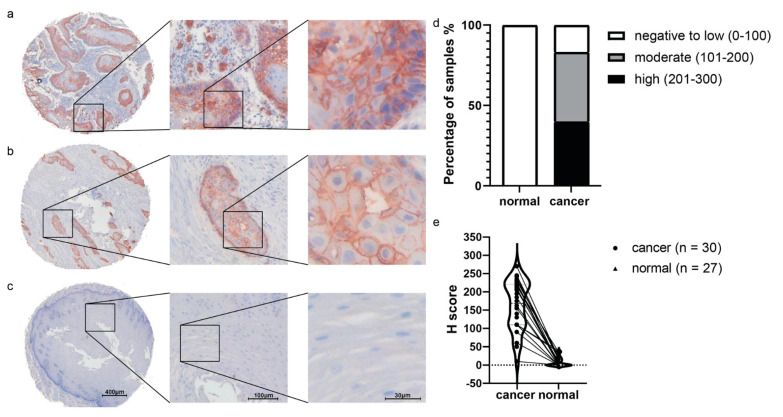
Anti-GLUT1 immunohistochemistry results in both ESCC and normal esophagus tissue. Representative images of anti-GLUT1 IHC staining results (red) in ESCC patient’s tissue ((**a**,**b**), different patients for each TMA displayed), and normal esophagus tissue (**c**). The images of (**a**–**c**) are from the same TMA slice. (**d**) Bar graph comparison results of H-Score in ESCC tissues to normal esophagus tissues, illustrating a significant difference in staining intensity (*p* value < 0.001). (**e**) Comparison of H-Score in ESCC tissues to normal esophagus tissues. Each line connects paired data from the same patient.

**Figure 4 ijms-22-09270-f004:**
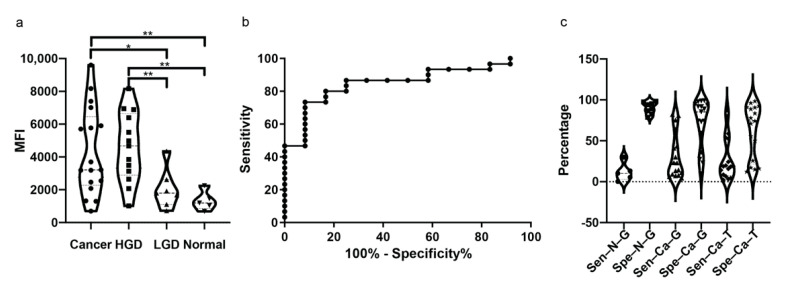
The fluorescent imaging results of 2-DG 800CW in patients’ biopsies. (**a**) The comparison results of the mean fluorescence intensity (MFI) in four different groups. The two-tailed Mann-Whitney U test is performed. * means *p* value < 0.05. ** means *p* value < 0.01. (**b**) The ROC curve analysis on the sensitivity and specificity of applying MFI in whole biopsy to distinguish cancerous tissue (including HGD and ESCC tissue) from LGD/normal tissue. (**c**) The sensitivity and specificity calculated on 4 μm slices. Sen–N–G or Spe–N–G means the sensitivity or specificity, respectively, of 2-DG 800CW to detect GLUT1 expression on normal and LGD tissue; Sen–Ca–G or Spe–Ca–G means the sensitivity or specificity, respectively, of 2-DG 800CW to detect GLUT1 expression on HGD and ESCC tissue; Sen–Ca–T or Spe–Ca-T means the sensitivity or specificity, respectively, of 2-DG 800CW to detect tumor area on HGD and ESCC tissue.

**Figure 5 ijms-22-09270-f005:**
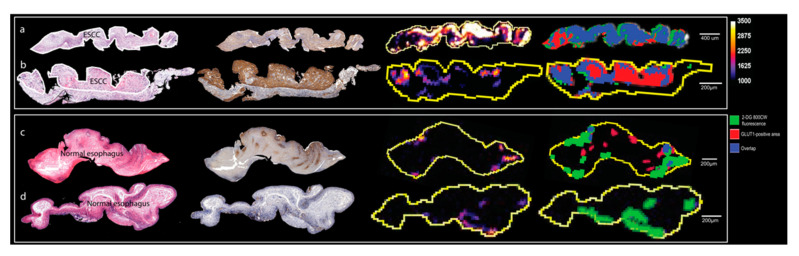
The merge percentage of 2-DG 800CW fluorescence signal in GLUT1 positive area. Each part (**a**–**d**) shows serial slices of one tissue block. The first column shows the H&E images, with cancer tissue delineated with white lines; the second column shows the anti-GLUT1 IHC results, with positive area stained in brown color; the third column shows the fluorescent images of 2-DG 800CW with a calibration bar of 1000 to 3500; the fourth column shows the merge of 2-DG 800CW fluorescent signal with GLUT1 positive area with a calibration bar of 1000 to 3500. Part (**a**,**b**) are ESCC tissue slices; (**c**,**d**) are normal esophagus tissue slices.

## Data Availability

The data presented in this study are available in Appendix A here.

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
