# Peer review of "Identification and Validation of Esophageal Squamous Cell Carcinoma Targets for Fluorescence Molecular Endoscopy"

_ijms, 2021, doi:10.3390/ijms22179270_

Round 1

Reviewer 1 Report

The manuscript titled "Identification and validation of esophageal squamous cell carcinoma targets for fluorescence molecular endoscopy" describes identification of esophageal squamous cell carcinoma markers that can help in the early detection of ESCC via fluorescence molecular endoscopy. The authors have used patient tumor and normal samples to compare the expression of GLUT1. The goal of the study, as stated in the abstract is to develop fluorescence molecular endoscopy to detect early mucosal lesions. The authors need to answer the following questions:

  1. Do the authors think that using only 1 surface target molecule, such as GLUT1, is sufficient or could the specificity for identifying early lesions be improved by combining it with another target such as VEGF?
  2. Trop2 is emerging as a specific specific marker in other cancers such as colorectal cancer. In this study, it ranked 337. A 2004 study by Nakashima et al. (DOI: 10.1002/ijc.20517) identified trop2 in sera of patients with ESCC. Could the authors compare the findings in that study with their own?
  3. Was there any difference between GLUT1 MFI for LGD vs. normal and HGD vs. cancer? Under discussion (lines 245-248), what are the authors referring to when they say "early-stage ESCC"? Do they mean high-grade dysplasia? If yes, then include this to make the results more clear.

Author Response

We would like to thank the Editors and the reviewers for their time and effort spent on reviewing our manuscript. We respond to their comments in a point-by-point manner and have adapted the manuscript accordingly.

Reviewer 2 Report

This paper is interesting but there are concepts that are not clearly expressed and some incorrect definitions.

Specifically, I would like to highlight:

1- Please, specify better histological definition of 'preinvasive lesions' with adequate reference (I suggest WHO classifications of Tumours of Digestive Sistem, 2019 edition);

2- There is a little confusion between 'preinvasive lesions' and 'early ESCC'. Please, clarify better in the whole text;

3- Immunohistochemistry is unable to detect genes; immunohistochemistry reveals proteins on cells. Moreover, there is a very confusing (awful) sentence: page 2, lines 93-94: '12 genes are localized on the cell membrane completely or predominantly': genes are located in the nucleo, not on cell membrane!!!!!!!!!!!!!!!!!!!!!!!!!!!!!!!!!!!!!!!!!!!

4- References of supplementary table 3 must be included in the main text.

Finally: who, among Authors, is the Pathologist who developed and carried out the immunohistochemistry, scored the results on tissue samples, indispensable for this work? I cannot find it!!! Please, clarify this point.

For details, please find enclosed peer-review text.

Author Response

(The authors gave the same response as above.)

Round 2

Reviewer 2 Report

The paper has been improved.

Well done.